# Nitric Oxide Acts as a Key Signaling Molecule in Plant Development under Stressful Conditions

**DOI:** 10.3390/ijms24054782

**Published:** 2023-03-01

**Authors:** Murtaza Khan, Sajid Ali, Tiba Nazar Ibrahim Al Azzawi, Byung-Wook Yun

**Affiliations:** 1Department of Horticulture and Life Science, Yeungnam University, Gyeongsan 38541, Republic of Korea; 2Department of Applied Biosciences, Kyungpook National University, Daegu 41566, Republic of Korea

**Keywords:** nitric oxide, antioxidants, phytohormones, signaling, stress, crop plants

## Abstract

Nitric oxide (NO), a colorless gaseous molecule, is a lipophilic free radical that easily diffuses through the plasma membrane. These characteristics make NO an ideal autocrine (i.e., within a single cell) and paracrine (i.e., between adjacent cells) signalling molecule. As a chemical messenger, NO plays a crucial role in plant growth, development, and responses to biotic and abiotic stresses. Furthermore, NO interacts with reactive oxygen species, antioxidants, melatonin, and hydrogen sulfide. It regulates gene expression, modulates phytohormones, and contributes to plant growth and defense mechanisms. In plants, NO is mainly produced via redox pathways. However, nitric oxide synthase, a key enzyme in NO production, has been poorly understood recently in both model and crop plants. In this review, we discuss the pivotal role of NO in signalling and chemical interactions as well as its involvement in the mitigation of biotic and abiotic stress conditions. In the current review, we have discussed various aspects of NO including its biosynthesis, interaction with reactive oxygen species (ROS), melatonin (MEL), hydrogen sulfide, enzymes, phytohormones, and its role in normal and stressful conditions.

## 1. Introduction

NO was first discovered in plants by Klepper [1]. However, the discovery of the critical functions of NO in mammals prompted its research in plants. In animals, Palmer et al. [2] investigated the role of NO as an endothelium-derived relaxing factor. This discovery grabbed the attention of scientists to further explore the role of NO as a chemical messenger in other living organisms under different environmental conditions. Early studies on NO focused on its environmental impact, since it is a significant pollutant. NO is abundantly produced in the atmosphere due to industrial combustion and automobile emissions, and it damages the ozone layer and causes acid rain [3]. Therefore, in the beginning, NO was considered a toxic gaseous molecule. However, later on, NO was explored as a significant signalling molecule in both plants and animals to regulate their physiological, molecular, and biochemical aspects under both normal and stressful conditions [4]. Further studies explored that NO is produced in living organisms via different enzymatic and nonenzymatic reactions and that NO can also be applied to plants in the form of different NO donors such as sodium nitroprusside (SNP) [5]. As a multifunctional molecule, in 1992, NO was regarded as a molecule of the year [6].

NO is a small, colorless, lipophilic free radical and a diatomic gas molecule that consists of nitrogen and oxygen. Therefore, NO can easily diffuse through cell membranes and can work as an excellent intracellular and intercellular chemical messenger in both plants and animals under normal and stressful conditions [4]. Under normal conditions, NO significantly contributes to the breaking of seed dormancy and induction of seed germination, root development, lateral root formation, primary root growth, adventitious root formation, root hair development, chlorophyll biosynthesis, vegetative growth, vascular differentiation, symbiosis nodule formation, stomatal moment, iron homeostasis, leaf senescence, flowering control, and fruit ripening [7]. Furthermore, NO also plays a pivotal role to protect plants from various environmental stresses, including wounding, pathogen invasion, hypoxia, ultra violet (UV) radiation, drought stress, salt stress, temperature extremes, and heavy metals stresses [5,8,9].

Compared with animals, NO production in plants is still a subject of controversy. In mammals, the main pathway responsible for NO production involves the conversion of L-arginine to citrulline by nitric oxide synthase (NOS). Although NOS-like activities in plants are susceptible to mammalian NOS inhibitors, no typical NOS sequence was discovered in the more than 1000 sequenced transcriptomes of terrestrial plants [10]. However, Foresi, et al. [11] reported the presence of NOS-like enzyme in green algae. In addition, in plants, NO is produced from different enzymatic and nonenzymatic routs, which are in continuous examination [12]. Cytosolic nitrate reductase (NR) has been considered as one of the main enzymatic sources of NO production in plants under aerobic conditions [13]. NR-dependent NO production pathway is involved in the production of NO during pathogen-induced defense responses, drought stress, cold stress, regulation of stomatal opening and closing, mitigation of symptoms caused by iron deficiency, and root-growth-associated responses [14]. Furthermore, NO can also be synthesized non-enzymatically from NO_2_^-^ in the presence of a reductant, such as ascorbate [15]. NO is produced in different plant organelles, including chloroplasts, mitochondria, peroxisomes, cytoplasm, cell wall, and cell membrane [16].

Although ROS are produced in the plant as a part of normal cellular metabolism, and at optimum concentrations can act as signaling molecules, if overproduced they can cause oxidative stress damages [4]. Plants primarily deal with the oxidative stress via an antioxidant defense system [4]. During abiotic stress, ROS are overproduced, resulting in an increased NO production which indicates the interaction between ROS and reactive nitrogen species (RNS). For instance, the application of hydrogen peroxide (H_2_O_2_) significantly enhanced the production of NO in *Arabidopsis*, which leads to the induction of antioxidants activities and reestablishment of redox balance [17]. On the other hand, NO production was reduced with the removal of H_2_O_2_ [18]. Furthermore, calcium ion channel’s inhibitors also inhibit H_2_O_2_-induced NO synthesis [19]). Similarly, H_2_O_2_ is crucial for the production of NO mediated by abscisic acid [20]. In contrast, the production of H_2_O_2_ mediated by ABA does not depend on NO [21]). Importantly, both H_2_O_2_ and NO are essential for signal transduction as well as phytotoxicity [22]. The NO and ROS signaling pathways in the defense system of plants are closely connected. During stress conditions, NO and ROS can alter gene transcription profiles and the production of plant hormones and metabolites. For instance, NO reacts with superoxide anion (O_2_^−^) to form peroxynitrite (ONOO^−^), which leads to post-translational modification (PTM). Several studies over the last two decades have demonstrated the involvement of NO and ROS in plant defense systems, where they show a symbiotic relationship [23]. Furthermore, in plants, melatonin also acts as a multipurpose molecule and regulates plant growth and development, and defense responses against biotic and abiotic stress conditions [24,25]. NO and melatonin (MEL) interact to induce plant growth and development under biotic and abiotic stress conditions [24]. Along with NO, MEL promotes plant growth and modulates the ability of plants to withstand abiotic stress by strengthening the antioxidant system [26]. MEL can induce NO synthesis by boosting the activities of NOS and scavenging NO if overproduced [27,28]. NO and MEL combine to form NOmela, a newly discovered chemical that may exert a more potent signaling effect than either MEL or NO alone [29]. In addition, melatonin also acts as a chemical scavenger for ROS and reactive nitrogen species (RNS) to improve the antioxidant system of the plants and protect the plants from oxidative damages [30].

Interactions between NO and antioxidants and plant hormones further broaden the signaling role of NO in plant growth and development under normal and stressful conditions. It has been reported that NO affects the activities of antioxidant enzymes, such as catalase (CAT), superoxide dismutase (SOD), various peroxiredoxins (Prxs) and enzymes involved in the ascorbate-glutathione (Asa-GSH) cycle, through PTMs [31]. In recent studies it has been reported that different NO donors such as SNP, S-nitrosocysteine (CySNO) and S-nitrosoglutathione (GSNO) are used to exogenously to supply NO to the plants to improve their defense system against different stresses [5,9]. Moreover, SNP has been widely used as a NO donor in pharmacologic studies to investigate its function in NO synthesis. While in plants, SNP has been used as a NO donor to elicit the synthesis of important bioactive secondary metabolites [32]. Application of SNP (a NO donor) on *Panax ginseng* and rice plants, significantly enhanced the activities of antioxidant enzymes such as CAT, SOD, peroxidase (POD), polyphenol peroxidase (PPO) and APX to protect the plants from the toxic effects of salt and lead stress [5,33]. NO also acts synergistically with auxin (AUX) to control a variety of plant responses, including root organogenesis, gravitropic responses, formation of root nodules, responses to iron deficiency, activation of cell division, formation of embryogenic cells, and stimulation of nitrate reductase (NR) activity [34]. Application of SNP and AUX significantly enhanced photosynthetic efficiency and antioxidant system of *Brassica juncea* plants under salt stress. Moreover, the combined application of SNP and AUX was more effective than their individual treatment [35]. However, AUX has been found to have little to no impact on the activation of NO production under specific experimental conditions or in specific cell types [36]. On the other hand, NO has been found to exert both synergistic and antagonistic effects on cytokinin (CK) metabolism. NO has been reported to promote the activation of CK to correct the meristematic abnormalities of the *noa1* mutant in root tissues [37]. Furthermore, high concentrations of NO are known to inhibit CK signaling [38]. Under stress conditions, NO interacts with ABA to induce the plant defense system [39]. The interaction between NO and ABA regulates crassulacean acid metabolism (CAM) expression [40], thereby aiding plant survival under drought stress and nutrient deficiency. NO-induced *AtAO3* and *AtNCED3* significantly improved drought stress tolerance in *Arabidopsis thaliana* [39]. NO interacts both synergistically and antagonistically with gibberellins (GAs) and has been reported to function upstream of GA to regulate its production, perception, and transduction [41]. In contrast, in one study, endogenous GA concentration was significantly reduced in *Arabidopsis* plants treated with SNP [41]. Most of the recent studies have shown a negative relationship between NO and ethylene (ETH). For instance, exogenous application of NO delays the senescence of both vegetative and reproductive organs by inhibiting factors involved in ETH generation [42].

The role of NO as a chemical messenger has been studied extensively with respect to plant growth and development under normal and stressful conditions. Pandey et al. [43] reported that during seed germination, NO suppresses cytochrome oxidase (COX) to increase respiration and thereby provide more energy for germination. Furthermore, NO has also been reported to significantly contribute to plant growth and development, symbiotic associations, defense responses against biotic and abiotic stressors, and vegetative and reproductive growth [4,7]. NO-induced *AtILL6* gene positively regulated plant’s basal defense, defense mediated by resistance (*R*) gene and systemic acquired resistance (SAR) against virulent and avirulent pathogenic bacteria in Arabidopsis thaliana [9]. Similar findings were also observed by Shahid, et al. [44], and they stated that NO-induced *AtCLV1* and *AtCLV2* positively regulated the plant’s immunity against pathogenic bacteria. In contrast, Nabi et al. [45] reported that NO-induced *AtBZIP62* negatively regulated plants’ basal defense and SAR in *A. thaliana*. However, the NO-induced *AtBZIP62* significantly enhanced the resistance of *Arabidopsis thaliana* against drought stress [8]. Furthermore, NO-induced *AtAO3* and *AtNCED3* negatively regulated immunity of *A. thaliana* but positively regulated the resistance of it to drought stress [39]. Application of NO-donors also significantly improved the defense system of different plants against abiotic stress conditions. For instance, the application of SNP significantly improved the defense system of rice against salinity and lead stress [5,46]. Similarly, SNP application increased the defense system of soybean plants against flooding stress [47].

## 2. NO Biosynthesis

NO was first studied by Joseph Priestley in 1772, who noted that NO was a colorless, toxic gas [6]. In 1879, William Murrell first used nitroglycerin to treat angina pectoris, but it was not until 1977 that Ferid Murad discovered that the release of NO causes nitroglycerin to exert positive pharmacological effects on vascular smooth muscle [6]. Furthermore, Palmer, Ferrige and Moncada [2] identified NO as an endothelial relaxation factor. Eventually, in 1992, NO was regarded as the “molecule of the year” [6]. In plants, NO was first discovered by Klepper [1]. In animals, NO is produced in the presence of NADPH and O_2_ by the conversion of L-arginine to N-hydroxyarginine via NOS, which also results in the production of L-citrulline. However, in higher plants, it is unknown whether NOS is present. The production of NO via NOS, nitrate/nitrite reductase, other enzymes, and non-enzymatic reaction pathways is discussed in the following subsections.

### 2.1. NO Production through NOS

NOS has not yet been discovered in higher plants, but has been found in *Ostreococcus tauri*, a unicellular alga. This suggests that higher plants may have evolved different mechanisms of NO production than those found in animals [48]. However, some studies have shown that NO production using NOS-associated pathways also exists in plants, since experiments conducted on maize, pea, and tobacco found that the use of NOS inhibitors and antibodies (monoclonal and polyclonal) reduced the production of NO [49,50]. Furthermore, it has also been demonstrated that the NO levels of Arabidopsis were raised in response to the expression of rat neuron-type NOS [51]. However, the genes and protein sequences of plant NOS-like enzymes are substantially dissimilar from those found in animals. For example, multiple NOS-like proteins in maize were found to show limited protein sequence homology with animal-derived NOS sequences [52]. Guo, Okamoto and Crawford [36] discovered that the protein produced by the *AtNOS1* gene has a similar sequence to one involved in NO generation in gastropods (snails), but that this differs from other animal NOS proteins. Later research by Moreau et al. [53] revealed that AtNOS1 is a circularly-permuted GTPase (cGTPase), which led to the renaming of this protein to AtNOA1. It may play a role in mitochondrial and ribosomal biosynthesis and translation and may act indirectly to regulate NO generation [53,54].

### 2.2. NO Production via the Nitrate or Nitrite Reductase and Other Enzymes

The cytoplasm within plant cells contains an enzyme called NR, which catalyzes the conversion of nitrate and nitrite to NO. This conversion is thought to be the main source of NO in plants [55,56]. A previous study reported that nitrite and endogenous NO levels were considerably lower in *Arabidopsis NR* deletion double mutant *atnia1*/*atnia2* relative to wild-type (WT) plants and concluded that the *NR* genes *AtNIA1* and *AtNIA2* are involved in NO production [57]. NR activity was subsequently detected in a variety of plants, such as maize, cucumber, sunflower, and spinach [56,58]. *NR* activity can be controlled through PTM of proteins, including phosphorylation, and via redox pathways; both forms of regulation can impact NO generation [59].

Research has demonstrated that the nitrite reductase (NiR) reaction pathway occurs in tobacco root cells [60]. The presence of high NO concentrations in transgenic tobacco plants containing an antisense version of the *NiR* gene suggests that *NiR* is involved in the generation of endogenous NO in tobacco plants [61]. Since NiR derived from plastids reduces nitrite to ammonia under normal circumstances, the nitrite content of plant cells is significantly lower than their nitrate content [56]. This is thought to inhibit NiR activity via competitive inhibition. As a result, when NADH and nitrate levels are sufficient, NR is much less effective at catalyzing the conversion of nitrite to NO than it is at catalyzing the conversion of nitrate to NO [56]. However, further research is needed to determine the exact molecular regulatory mechanism by which NR effectively converts nitrite to NO.

In addition, NR can function as a cofactor to help other enzymes produce NO. For instance, in a unicellular algae (*Chlamydomonas reinhardtii*), NR was observed to help the amidoxime-reducing component (ARC) to convert nitrite to NO and replace the electron transfer functions of cytochrome b5 (cytb5) and cytochrome b5 reductase (cytb5R) [62]. The major function of plant ARC proteins may be the manufacturing of NO, which is why they are also known as NO-forming nitrite reductases. The *Arabidopsis* genome has two ARC genes that show a high degree of evolutionary conservation and are similar to those found in *C. rhinoceros* [63].

Numerous enzymes, such as horseradish peroxidase (HPOX) [64], xanthine oxidoreductase (XOR), xanthine dehydrogenase (XDH) [65], cyt P450 [66], polyamine oxidase (PAO), polyamine oxidoreductase (POR) [48], and copperamine oxidase (CuAO) [41], also produce NO in plant cells. For instance, PAO uses polyamine (PA) and POR uses hydroxylamine (HA) as their respective substrates for the production of NO. CuAO in *Arabidopsis* has been found to increase arginine and NO levels by decreasing the activity of arginase [67]. However, further investigation of proteins or protein complexes in plants associated with arginine-dependent NO generation is warranted. Several molybdoenzymes, including xanthine oxidase (XO), aldehyde oxidase (AO), and sulfite oxidase (SO), also reduce nitrite to NO under low oxygen conditions [68], although the precise reduction pathways involved and their associated molecular regulatory mechanisms remain poorly understood.

### 2.3. NO Production via Nonenzymatic Reaction Pathways

It has been demonstrated that NO can also be formed nonenzymatically by the reduction of nitrite in the presence of antioxidants, acidic and reducing agents, or as a byproduct of direct chemical interactions between nitrogen oxides and plant metabolites [69]. Exogenous nitrite treatments used in earlier studies were found to partially restore NO levels and *Pseudomonas syringae* resistance in *Arabidopsis nia1/nia2* mutants. This raises the possibility that plants may have a mechanism for producing NO that is totally independent of enzymatic processes [59,70]. For example, Bethke et al., found that following exogenous application of nitrite, the ectoderm of barley pasteurized cells underwent a non-enzymatic reaction to reduce nitrite to NO [71]. Furthermore, some arginine metabolites, including PAs and HAs, can be used by plants to produce NO [68]. However, further research is required to fully understand the physiological and biochemical functions of these pathways and their underlying molecular mechanisms. A summary of different sources and pathways related to NO production discussed above is shown in Figure 1.

## 3. NO Interactions with ROS, Melatonin, and Hydrogen Sulfide (H_2_S)

Considerable progress in elucidating the intricate nature of NO and ROS interactions has been made in the last two decades. Both NO and ROS play significant roles as biological messengers in response to biotic and abiotic stressors. In addition to controlling plant growth and a variety of physiological responses to the environment, including germination, root growth, gravitropism, and stomatal closure, NO and ROS interactions are crucial [72]. The overproduction of ROS under abiotic stress leads to increase NO production, mainly via the activity of NR, which suggests a link between ROS and reactive nitrogen species (RNS) [17]. In this study, exogenous H_2_O_2_ administration to *Arabidopsis* plants caused an 8-fold increase in NO formation, and subsequent NO accumulation activated antioxidant defense mechanisms, reduced ROS overproduction, and restored redox balance [17]. However, it is also known that NO generation can be arrested by the elimination of H_2_O_2_ by antioxidants or NADPH oxidase inhibitors [18]. Moreover, NO generation mediated by ABA requires H_2_O_2_ [20]. In contrast, treatment with a NO donor (e.g., SNP), a NO scavenger, and an inhibitor of NO synthesis revealed that ABA-induced H_2_O_2_ generation does not depend on NO. As a result, H_2_O_2_ can actively control NR activity to influence NO production [21]. Furthermore, H_2_O_2_ and NO are essential for signal transduction despite being phytotoxic [22]. Numerous ROS and RNS, including singlet oxygen (^1^O_2_) and peroxynitrite (ONOO^-^), may develop in response to abiotic stress and result in oxidative damage [73]. On the other hand, superoxide anion (O_2_^−^) and NO react to form ONOO^−^, a potent oxidant that is involved in the tyrosine nitration PTM of proteins [74,75]. In peroxisomes, S-nitrosylation prevents the activity of glycolate oxidase and CAT and regulates H_2_O_2_ levels at the cellular level [76]. In contrast, ONOO^−^ causes tyrosine nitration and nitrosative stress in plants. Furthermore, a proteomic analysis revealed that peroxisomal NADH-dependent hydroxy pyruvate reductase can become dysfunctional in response to nitration by peroxynitrite [77]. It can, therefore, be concluded that the interaction between ROS and NO causes several harmful or signaling effects in which many other elements are involved.

Recent studies have highlighted the discovery of a fascinating relationship between MEL and NO. For example, Wen et al. [78] demonstrated that the application of MEL to tomato plants stimulated NO synthesis by encouraging NR activity. This, in turn, induced the formation of adventitious roots via modulation of the expression of AUX-related genes, including those involved in AUX accumulation, transport, and signal transduction. In addition, MEL can control the NO/NOS system to carry out physiological tasks [79]. Similarly, NO exogenous administration has also been shown to stimulate MEL synthesis in tomato seedlings [80]. This was found to induce root development and, therefore, suggests that NO and MEL may act in a feedback loop that affects root development via AUX signaling pathways. In addition, it has been demonstrated that MEL reduces the negative effects of stress by raising NO levels. This occurs as a result of the elevation of NR and NO synthase-related activities and of the coordination of the polyamine pathway [81]. By directly scavenging free radicals, NO and MEL operate as antioxidants, reducing oxidative damage independently of receptors [82]. In addition, interactions between NO and MEL are thought to be essential for inducing the appropriate PTM of important stress-related proteins; this hypothesis has been validated by proteomics research [83]. For example, in sunflower, NO was found to be able to reduce growth inhibition caused by salt stress by inducing MEL accumulation and subsequently triggering modulation of the expression of Cu/Zn SOD and MnSOD [84].

Like ROS, NO, and MEL, hydrogen sulfide is an active gas molecule that significantly contributes to plant growth, development, and defensive responses to biotic and abiotic stress [84]. Moreover, each of these chemicals can protect against or promote damage to plant cells (synergistically or antagonistically) depending on their relative quantities. However, recent evidence demonstrates that intricate biological relationships between NO and H_2_S include numerous pathways that differ between plant organs, species, and environment or experimental setup [85]. For example, one study showed that non-climacteric *Capsicum annuum* ripening is associated with a decrease in NO content and an increase in H_2_S content [86]. Both NO and H_2_S play crucial signaling roles under normal and adverse conditions via transcript accumulation, PTMs, and interactions with other plant regulators [85]. For instance, H_2_S mediates persulfidation and NO mediates S-nitrosation, both of which are important PTMs [85]. One key target of S-nitrosation and persulfidation is CAT, an important antioxidant enzyme that is only found in peroxisomes [87,88]. H_2_S can also interact with other oxidizing agents, including hydrogen peroxide (H_2_O_2_), oxygen (O_2_), hydroxyl radical (OH^−^), and ONOO^−^. In *Arabidopsis* and pea, NO and H_2_S both positively control APX activity, with the Cys32 residue being the target of both of their respective PTMs. This finding suggests that APX activity is tightly controlled and that there is a delicate equilibrium state for H_2_O_2_, NO, and H_2_S. However, the interactions between these molecules and the intricate system of plant metabolism [89] requires more in-depth research.

## 4. Interactions between NO and Enzymes

The combination of NO and enzymes including antioxidant compounds further broadens the signaling capabilities of NO. Unlike NO, nitrosating molecules, such as NO+, N_2_O_3_, and S-nitrosothiols, react with reduced ascorbate [90]. As a result, NO is discharged and ascorbate (AsA) is transformed into dehydroascorbate (DHA) [91]. When NO and the ascorbyl radical are combined, DHA spontaneously degrades to O-nitrosoascorbate. The latter is then hydrolyzed to produce ascorbate and nitrite (NO^−^_2_) [90]. In addition, AsA can scavenge ONOO^−^ to produce NO^−^_2_ and nitrate (NO^−^_3_) via unidentified intermediates; this reaction proceeds slowly at neutral pH but is much quicker at pH 5.8 [92]. Similar to how NO^−^_2_ interacts with the glutathiyl radical to create nitroglutathione (GSNO_2_), which then releases NO, GSH impacts ONOO^−^ levels by either reducing NO to NO^−^_2_ or via radical–radical interactions [93]. Further investigation into the biological relevance of ONOO^−^ breakdown mechanisms is required. However, in normal situations, high concentrations of GSH and AsA in plant cells might help maintain low levels of NO derivatives. Gamma-tocopherol [94]; carotenoids; and the flavonoids ebselen, epicatechin, and quercetin are other well-established scavengers of ONOO^−^ found in plants [95]. Some of these compounds not only scavenge ONOO^−^ but also NO and ROS.

NO can interact with glutathione through several routes. In the roots of *Medicago truncatula*, total glutathione levels were observed to increase following transcriptional activation of SNP and GSNO-stimulated genes involved in GSH production [96]. Moreover, it has been demonstrated that NO can affect the activity of main antioxidant enzymes found in plants, including CAT, SOD, peroxiredoxins (Prx), and the enzymes involved in the ascorbate-glutathione (Asa-GSH) cycle, to varying degrees via PTMs [31]. In Arabidopsis cells without the application of a NO donor, a total of 53 endogenous nitrosocysteine were found in proteins from all cell membranes and regions, serving a wide range of functions [97]. After exposure of Arabidopsis to cold stress Puyaubert et al. [98] found 62 endogenously nitrosylated peptides, of which 20 were over-nitrosylated. NO induces PTMs of several enzymes in different plants through the process of S-nitrosation or tyrosine nitration and these enzymes include 3-ketoacyl-Co thiolase 1, hydroxy pyruvate reductase, glycolate oxidase, CAT, MDAR, CuZnSOD etc. and are shown in Table 1.

## 5. Interactions between NO and Phytohormones

### 5.1. Interactions between NO and Auxins

AUXs are essential for controlling a variety of developmental processes under both normal and stressful conditions. A sufficient amount of indole-3-acetic acid (IAA) is required for the formation, development, and maintenance of plant roots [104]. In addition, indole-3-butyric acid (IBA), a naturally occurring IAA precursor, must be converted into IAA in order to maintain optimal IAA levels to support root system development [105]. AUX and NO have been found to work in concert to control a variety of plant responses, including root organogenesis, gravitropic responses, formation of root nodules, responses to iron deficiency, activation of cell division, formation of embryogenic cells, and stimulation of NR activity [34]. Increased NO production has frequently been observed following exogenous AUX administration or in AUX overproducer mutant lines [106]. In addition, NO is also produced when IBA converts into IAA [107]. In contrast, research in *Arabidopsis* using SNP or under stress conditions has demonstrated that NO can modulate auxin levels by affecting biosynthesis, degradation, conjugation, distribution, signalling, and enhancing auxin-influx by increasing the expression of the *AUXIN-RESISTANT 1* (*AUX1*) gene [108]. NO has been shown to influence rice root growth by controlling auxin transport under nitrate supply [109]. The auxin–NO interaction has also been the subject of numerous research under metal-stress circumstances; however, the outcomes are conflicting. Root meristem growth was significantly impacted by the Cd-induced NO-accumulation in the primary roots of *A. thaliana*, which decreased auxin levels and negatively regulated auxin transport and signalling [110]. Exogenously applied NO donor improved rice resistance against mercury stress by enhancing the transport of IAA in roots [111]. However, in response to iron deficiency, NO inhibited root elongation by decreasing AUX levels [112]. In rice, arsenic and cadmium stress adversely affects the root system by changing the IAA production and distribution; however, NO application reduces the toxic effects of cadmium stress [113,114]. Elimination of NO has been found to greatly reduce typical AUX-dependent root responses, including the activation of mitogen-activated protein kinases (MAPKs) during adventitious root formation and the stimulation of cell cycle genes during lateral root formation [115,116]. Similarly, a strong link between asymmetric AUX distribution and endogenous NO localization was observed during gravitropic bending in soybean roots and during the formation of indeterminate nodules in the roots of Medicago species infected by AUX-overproducing rhizobia [117,118].

Although the effect of AUX on NO generation is well-established, recent research has also shown that NO may also influence AUX metabolism, transport, and signaling. For instance, it has been shown that NO increases root indole-3-acetic acid (IAA) concentrations in *Medicago truncatula* seedlings exposed to cadmium (Cd) stress. It does so by reducing the rate of IAA breakdown via the activity of IAA oxidase, thus increasing AUX availability and reducing Cd stress [114,119]. In addition, experiments using pharmacological interventions and/or mutants to produce excessive NO showed that high concentrations of NO inhibit acropetal AUX transport in the roots of Arabidopsis plants by decreasing the abundance of the AUX efflux protein PIN-FORMED 1 (PIN1) via a proteasome-independent post-transcriptional mechanism [120]. Caused by a reduction in cell division and the promotion of cell differentiation, this NO-dependent decrease in PIN1 protein levels and the resulting disruption of root AUX transport severely reduced root meristem size and activity in primary roots, impairing the maintenance of the root apical meristem and primary root growth [120]. Furthermore, the AUX receptor protein TIR1 (TRANSPORT INHIBITOR RESPONSE 1) is S-nitrosylated at two specific cysteine residues, suggesting that NO directly influences AUX perception and signal transduction [121]. The S-nitrosylation of TIR1 appears to facilitate TIR1′s interaction with AUX/INDOLE-3-ACETIC ACID (AUX/IAA) proteins, which operate as transcriptional repressors of genes related to the AUX response. Thus, to inhibit the expression of AUX-dependent genes, TIR1 targets AUX/IAA proteins for proteasome degradation. Due to the enhanced TIR1-AUX/IAA connection caused by TIR1 S-nitrosylation, AUX-dependent gene expression may be promoted by facilitating AUX/IAA degradation through the proteasome [121]. This NO-dependent PTM may also affect TIR1′s ability to bind AUX, although further research on this topic is required.

### 5.2. Interactions between NO and Cytokinins

NO has been found to exert both synergistic and antagonistic effects on cytokinin (CK) metabolism. As a consequence, NO can function downstream, mediating CK responses [122], or independently further upstream [123]; its function most likely depends on the condition of the plant cell. Surprisingly, most of the synergistic effects between NO and CKs arise from the combination of CKs and molecules that lower NO levels, including NO-scavengers, arginine-based NOS inhibitors, and/or NO-deficient mutants. For instance, when exposed to CKs, the NO-deficient mutant *noa1* showed decreased sensitivity to root growth suppression. In addition, NO can also promote the activation of CYCD3;1 by CKs during cell proliferation; CYCD3;1 overexpression has also been found to correct meristematic abnormalities found in the *noa1* mutant in root tissues [37]. The ability of NO to affect the hormonal system of wheat plants by increasing the concentration of hormones of a cytokinin nature under normal conditions and preventing a decrease in their level under stress is associated with its ability to realize the growth-stimulating and protective effects of NO [124].

However, by combining CKs with high NO levels (i.e., via NO donors or overexpression mutants), researchers have demonstrated that NO and CKs can exert antagonistic effects. In the *nox1*/*cue1* mutant, CKs reverse the modest shoot growth phenotype brought on by high endogenous NO levels [38]. In contrast, treatment with NO was found to result in a net reduction in CK activity, whereas exogenous CKs prevent the formation of primary roots. Moreover, high concentrations of endogenous NO (i.e., caused by the presence of excess SNP or GSNO, or in the *gsnor1* or *nox1* mutants) are known to inhibit CK signaling. The *gsnor1* and *nox1* mutants have also been found to show a lower sensitivity to CK’s ability to restrict root growth after supplementation with CK. However, further investigation is required to improve our understanding of the interactions between NO and CK.

### 5.3. Interactions between NO and Abscisic Acid

ABA is widely known for inhibiting seed germination and promoting seed dormancy during embryo maturation, which acts as a protective mechanism in unfavorable environment [125]. NO, on the other hand, has the opposite impact since it induces seed germination by breaking seed’s dormancy [126]. Moreover, the seeds which were treated with NO donors revealed lower ABA contents [127]. Seeds from *Arabidopsis* NO-deficient double *nia1nia* and triple *nia1nia2noa1-2* mutants are more dormant than wild-type seeds and are also sensitive to ABA channel suppression of germination, indicating the involvement of NO in seed dormancy breakdown [128]. Furthermore, S-nitrosation at Cys-153 results in the degradation of abscisic acid insensitive 5 (ABI5), which encourages seed germination and seedling growth [129].

Stomatal mobility, which is principally controlled by a complicated ABA signalling network, is the mechanism by which gases are exchanged for plant photosynthesis [130]. In guard cells, ABA increases NO synthesis for cGMP and cADPR activity, which causes stomatal closure [131]. In the control of stomatal closure in *A. thaliana* guard cells, ABA exhibits a strong correlation with endogenous H_2_O_2_ and NO. To control stomatal mobility, ABA promotes the formation of endogenous H_2_O_2_, which raises the production of NO [18]. By using NO scavengers (cPTIO), NO donors (SNP), and inhibitors of the activities of L-arginine dependent NOS-like and NR, it has been established that ABA causes a crucial increase in NO content in guard cells [132].

Leaf senescence and fruit ripening are physiologically programmable processes that are characterised by a variety of phenotypical and biochemical alterations linked to an active ROS metabolism [133]. Both NO and ABA function as regulators of this ROS metabolism. During natural leaf senescence the production of NO is decreased. Therefore, the exogenous application of NO donors counteracts leaf senescence in rice by inducing ABA [134]. Furthermore, application of NO donors also delayed fruit ripening by the activation of antioxidant enzymes [133].

NO and ABA, two significant stress-related chemicals, actively interact via specific signaling cascades brought on by environmental stressors, which eventually induce plant adaptive responses [39,135]. In response to abiotic stress conditions, NO is produced in plants, which then act as a chemical messenger to enhance the production of plant hormones such as ABA, IAA, JA, SA, ETH and CK, antioxidant enzymes activities, chlorophyll biosynthesis, gene expression of antioxidant enzymes, and stomatal closure [136]. Furthermore, in response to biotic and abiotic stress conditions, NO modulate ABA pathways and regulatory networks in plants. Exogenous NO administration also stimulates the production of cGMP and MAPK activity, both of which are important mediators of ABA-induced stomatal closure, which lowers transpiration rate [137].

The *Arabidopsis nia1/nia2* knockout mutant with unchanged stomatal activity demonstrated that NO plays a function in stomatal closure triggered by ABA [137]. Similarly, type 2C protein phosphatases (PP2Cs), which function as negative regulators of ABA signaling, have also been hypothesized to play a role as putative crosstalk elements between ABA receptors and NO-mediated ABA signal transduction, possibly acting downstream of NO in intricate networks that regulate ABA-triggered stomatal closure [138]. NO and ABA also appear to interact closely to regulate CAM expression [40], which helps plants survive under water and nutrient-restricted conditions. NR activity appears to be the primary source of ABA-induced NO generation, and is associated with the control of stomatal movements in *Arabidopsis* as well as the regulation of CAM expression in bromeliads [138]. Liu et al. [139] also found that during the breaking of *Arabidopsis* seed dormancy, a rapid accumulation of NO in the endosperm preceded a decrease in ABA concentration. Moreover, they found that this pattern was connected to a sharp increase in the transcript and protein levels of the ABA 8′-hydroxylase CYP707A2, a crucial enzyme in ABA catabolism. It is, therefore, crucial to determine whether NO affects ABA receptors directly or is a downstream component in the ABA signaling cascade.

### 5.4. Interactions between NO and Gibberellins

Similar to NO, GAs play a pivotal role in plant growth and development. They affect numerous developmental processes, including seed germination, hypocotyl elongation, the acquisition of photomorphogenic traits, primary root growth, reorientation, and the growth of pollen tubes. However, actual interactions between NO and GAs have been described for only a small subset of physiological events related to these processes. In fact, most of the currently-understood mechanisms of how NO and GAs interact involve the control of seed germination [71,140] and the inhibition of hypocotyl elongation during seedling de-etiolation [41]. NO functions upstream of GA during the regulation of these responses, controlling GA production, perception, and transduction [71]. For most physiological processes in which both signaling molecules take part, there is evidence of an antagonistic relationship between them. A growing body of evidence also suggests that DELLA proteins play a crucial role in the crosstalk between GA and NO signaling pathways [41]. DELLA proteins, a relatively small family of transcriptional regulators, play a key role in the integration of different types of hormonal signals, including GA, ETH, jasmonate (JA), and ABA [141]. For example, during GA signaling transduction, hormone molecules connect to GA INSENSITIVE DWARF1 (GID1) receptors, which then bind a DELLA protein and drive the GA-GID1-DELLA complex to the E3 ubiquitin ligase SLEEPY1 (SLY1) [41]. This promotes DELLA destruction in the proteasome. However, new research has shown that NO exerts the opposite effect on cellular DELLA levels, causing their accumulation and negatively affecting GA signal transmission. Essentially, NO-driven DELLA accumulation can be explained as a decrease in tissue sensitivity to GA. This is because more GA-GID1-DELLA complexes are required to mark DELLA proteins for proteasome degradation to achieve sufficient transcriptional de-repression of GA-regulated genes. The difference in how NO and GA affect DELLA regulation may help to explain, at least in part, their antagonistic interactions in the regulation of key physiological processes in Arabidopsis, including hypocotyl elongation [41] and primary root growth [142]. Exogenous NO has also been shown to cause the accumulation of GA-regulated DELLA proteins [41], most likely by adversely affecting the GID1-SLY1 system of DELLA tagging for degradation. This finding emphasizes the potential role that DELLAs may play in antagonistic interactions between GA and NO. However, as noted previously [41], DELLA-independent processes may also be involved, and this means that controlling DELLA turnover and activity may not be the primary target of NO action in controlling plant growth and other GA-mediated developmental responses.

Along with the detrimental effects on the GA signaling network induced by NO, it has also been hypothesized that these molecules have a mutually antagonistic relationship toward their own endogenous levels [41]. This theory is supported by the observation that etiolated seedlings of the GA-deficient *Arabidopsis* mutant *ga1-3* showed NO levels that were noticeably greater than those seen in the WT genotype. In addition, both the *ga1-3* mutant and WT seedlings displayed lower NO levels following GA3 administration, indicating that exogenous GA negatively influences NO production [41]. On the other hand, endogenous GA levels were found to be significantly lower in WT *Arabidopsis* plants treated with SNP [41]. A thorough examination of the expression of GA biosynthesis genes in *Arabidopsis* (i.e., GA20oxidase and GA3oxidase) and catabolism (i.e., GA2oxidase) revealed that only *GA20ox3* was significantly upregulated in the NO-deficient *nia1*,*2noa1-2* mutant and down-regulated in NO-treated WT seedlings [41].

However, in some situations NO appears to have a stimulatory rather than an inhibitory effect on GA biosynthetic machinery [71]. Bethke, Libourel, Aoyama, Chung, Still and Jones [71] noted that a synergistic interaction was necessary for the transcription of two GA3oxidase genes (i.e., *GA3ox1* and *GA3ox2*) after the breaking of dormancy in Arabidopsis seeds. Research on wheat roots, where SNP-induced apical development was linked to higher GA3 levels, provides another example of the beneficial relationship between GA and NO [143].

Regardless of whether or not NO and GA share a common signaling pathway cascade during seed dormancy breaking, the stimulation of seed germination by either of these substances can be blocked by sufficiently high concentrations of ABA [144,145]. In the case of seed dormancy breaking, it is almost certain that both molecules encourage germination in a variety of species [146], but it is not clear whether NO and GA interact during this process, or how they may do so. These questions require further clarification in future research.

### 5.5. Interactions between NO and Ethylene

Many studies have provided evidence of an antagonistic association between NO and ethylene (ETH). For instance, fruit ripening and the management of leaf and floral senescence are regulated by NO, which operates as an ETH antagonist [147]. Another study showed that exogenous administration of NO, whether by direct fumigation or by addition of compounds that release NO, delayed the senescence of both vegetative and reproductive organs by inhibiting factors promoting the generation of ETH [42]. Measurements of ETH and NO emissions during fruit ripening or plant senescence were consistent with this finding, and revealed opposite trends for these gases, in which ETH production rises and NO levels fall during the induction and establishment of ripening and senescence [148]. Manjunatha, Lokesh and Neelwarne [147], suggested that decreases in the transcription and/or activity of crucial ETH biosynthesis enzymes may cause NO to limit the formation of ETH in fruit. The production of ETH in vegetative and reproductive plant tissues depends on the activity of the enzyme ACC synthase (ACS), which transforms the immediate ETH precursor 1-aminocyclopropane 1-carboxylic acid (ACC) into S-adenosyl methionine. According to studies of climacteric fruits, exogenous NO can alter ACS and ACC oxidase (ACO) activity and transcription, which in turn affects ACC accumulation and the levels of ETH production [147]. Since the concentrations of ACC, ACS, and ACO in plant tissues play a key role in determining the rate at which ETH is produced, an inhibitory impact of NO on any of these substances likely reduces ETH synthesis [146].

### 5.6. Interaction between NO and SA

NO and SA are crucial signalling compounds which significantly contribute in plant growth and development and defense responses to biotic and abiotic stress conditions [34,149]. In plants, the application of both NO and SA significantly enhanced photosynthesis, plant water regulation, and eventually crop yield [149]. These findings reveal that both NO and SA are playing their roles in plant growth and development under normal conditions. Further findings also revealed that both NO and SA also significantly improve the defense system of the plants against biotic and abiotic stresses [9,149]. For example, Basalah et al. [150] reported that both NO and SA significantly mitigated the adverse effects of cadmium (Cd) on wheat seedlings. Similarly, Mostofa et al. [151] also reported that both NO and SA significantly enhanced the defense system of rice against Cd stress. The same authors further observed that the combined effects of both NO and SA were more effective than their sole application in the mitigation of Cd stress in rice. This study shows that SA and NO work together to reduce the negative effects of Cd on rice and possibly other crop plants. To date, numerous studies report that exogenously applied NO and SA significantly reduced the adverse and toxic effect of temperature extremes, drought, salinity, and heavy metals stress on different plants [5,149]. Furthermore, several studies also showed that both NO and SA significantly contributes in the induction of different modes of plant immunity including pathogen triggered immunity (PTI), effector triggered immunity (ETI) and systemic acquired resistance (SAR) against virulent and avirulent bacteria [44,149]. For instance, in *A. thaliana*, *IAA-leucine resistant (ILR)-like gene 6* (*AtILL6*) induced by NO and the *salicylic acid induction deficient 2* (*AtSID2*) positively regulated PTI, ETI and SAR, when the plants were exposed to virulent and avirulent pathogenic bacteria [9]. The author sated that NO-induced *AtILL6* and *AtSID2* are involved in the induction of plant immunity against pathogens.

## 6. Role of NO in Plant Growth and Development under Normal and Stressful Conditions

To maximize growth and crop yield, seed germination is tightly controlled by a mix of external factors and internal signals [150,151,152,153,154,155,156]. In agricultural production, a number of nitrogenous substances, including nitrate, nitrite and SNP, have been utilized to induce seed germination and seedling vigor [110,124,157,158,159,160,161,162,163]. According to further research, these nitrogenous compounds likely promote dormant seed germination by generating NO. Furthermore, NO also protects the germination of various plants under stress conditions, and presoaking attenuated the inhibition of seed germination and early growth of cucumber and *Brassica chinensis* under saline stress [164,165]. Furthermore, NO significantly increase germination percentage, seedling growth, biomass accumulation, yield in a number of vegetables, flowers and fleshy fruits, have been reported after the application of NO donors and symbiotic association with the microorganisms [34,135]. In addition, NO regulates cell processes in a variety of ways. NO-mediated suppression of cytochrome oxidase (COX) during seed germination enables fine-tuning of respiration to prevent tissues from becoming anoxic [43]. This minimizes ROS production, DNA damage, and lipid degradation. In addition, NO stimulates the expression of genes related to the cell cycle and raises the intracellular concentrations of amino acids, which are essential for protein synthesis during seed germination. Exogenous application of SNP, an NO donor, was found to significantly improve key agronomic traits of rice, including shoot length and number of tillers per plant [5]. Furthermore, NO, along with hormones and H_2_S, significantly contributes to fruit ripening. The application of NO delays ripening, minimizes chilling damage, improves immunity, and increases nutritional value, because it is known that the NO level gradually declines during ripening with concurrent protein nitration and nitrosation [166]. For example, application of NO was found to induce fruit maturation in sweet pepper [166]. This treatment reduced ROS production and increased antioxidant activity, thereby reducing oxidative damage. Furthermore, NO significantly contributes to senescence-related processes in plant tissues. This contribution might have either a positive or negative effect depending on a specific organ. For example, NO delays senescence in leaves but promotes senescence in root nodules [167].

Environmental stressors are harmful to plant growth. Plants have evolved mechanisms to deal with various forms of stress and balance important regulators of growth and stress. NO is a signaling molecule that is capable of significantly regulating the switch between development and stress responses. This occurs primarily by protein PTMs, including S-nitrosation, metal-nitrosylation, and Tyr-nitration [34,168]. To date, hundreds of S-nitrosated proteins have been discovered in plants in response to diverse biotic and abiotic stressors, and S-nitrosation has been described as a key mechanism regulating stress signaling [4,169]. It has also been shown that NO and hormones, such as ABA, are involved in crosstalking during stress signaling. This crosstalk involves the activation of transcription factors, which induce transcript accumulation in response to a variety of stressors [170]. For example, NO-induced genes from ABA pathways, such as *Arabidopsis thaliana aldehyde oxidase 3* (*AtAO3*), have been found to positively regulate drought stress in *Arabidopsis* [39]. Exogenous NO treatment has also been demonstrated to improve plant responses to flood stress, heavy metal stress, and deficits in critical mineral elements [5,47,171]. Moreover, in response to nutrient shortages, NO encourages root growth and triggers ion translocation mechanisms that promote mineral nutrient intake [172]. Under heavy metal stress conditions, the cytoprotective effects of NO are mediated by the activation of antioxidant defenses and consequent reduction of excessive ROS and oxidative cell damage [5,173].

Since early studies first reported the role of NO in aiding plant pathogen defense mechanism [174], NO has been found to be part of a network of host–pathogen elicitory events that regulate responses to pathogen attack. For example, tobacco plants exposed to the bacterial pathogen *Pseudomonas syringae* were used by Mur et al. [175] to show how both NO and nitrite can affect primary metabolic functions during the hypersensitive response. Moreover, the addition of 1 mM CySNO to *Arabidopsis thaliana* plants resulted in the induction of thousands of genes related to plant growth and development or to defense responses against biotic and abiotic stresses [176]. The roles that these genes play have been investigated with respect to plant growth traits and responses to both biotic and abiotic stress conditions [8,9,44,45]. In addition, it has been demonstrated that NO plays a role in the regulation of plant defenses against pathogenic fungi [177]. However, NO is also produced by the plant in order to recognize symbiotic fungi, and it is especially important for optimal control of mycorrhizal symbiosis [34,178]. Taken together, as a signaling molecule, NO significantly contributes to break seed dormancy and induce seed germination. Moreover, it is also linked to regulation of the root system, photosynthetic pigments, and overall vegetative growth [7]. Furthermore, NO also plays a pivotal role in the induction and establishment of symbiotic association [34]. NO also significantly contributes to plant reproduction [179]. The pivotal role of NO in plant growth and development, symbiotic association, vegetative growth, reproductive growth, and defense responses against biotic and abiotic stressors is illustrated in Figure 2.

## 7. Conclusions

Nitric oxide is a signaling molecule that exerts key functions at the physiological, biochemical, and molecular levels. Environmental stresses, especially in early plant growth and development, pose a considerable threat to plant growth and productivity. Both biotic and abiotic forms of stress damage plants and subsequently cause a reduction in plant growth attributes and yield. However, NO has been found to efficiently alleviate the negative effects of both biotic and abiotic stresses in plants. The key function of NO is based on its concentration, signaling and interactions with other molecules. Interactions between NO and phytohormones, MEL, H_2_S, and ROS modulate plant functions under stressful conditions. Moreover, crosstalk between NO and other molecules enhance plant tolerance to both biotic and abiotic stress conditions, but the mechanism underlying this phenomenon requires further detailed research. We also highlighted the signaling of NO in plants and its interactions with other important biochemicals, as well as discussed the role of NO in mitigating biotic and abiotic stresses.

## Figures and Tables

**Figure 1 ijms-24-04782-f001:**
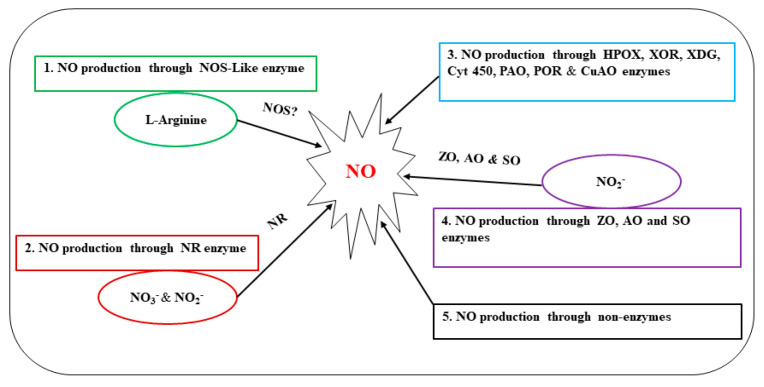
Production of NO through different pathways. Nitric oxide synthase (NOS), not discovered in higher plants (NOS?), Nitrogen oxide (NO), nitric oxide synthase (NOS), nitrate reductase (NR), nitrite ion (NO_3_^−^), nitrate ion (NO_2_^−^), horseradish peroxidase (HPOX), xanthine oxidoreductase (XOR), xanthine dehydrogenase (XDG), cytochrome (cyt P450), polyamine oxidase (PAO), polyamine oxidoreductase (POR), copperamine oxidase (CuAO), xanthine oxidase (ZO), aldehyde oxidase (AO), and sulfite oxidase (SO).

**Figure 2 ijms-24-04782-f002:**
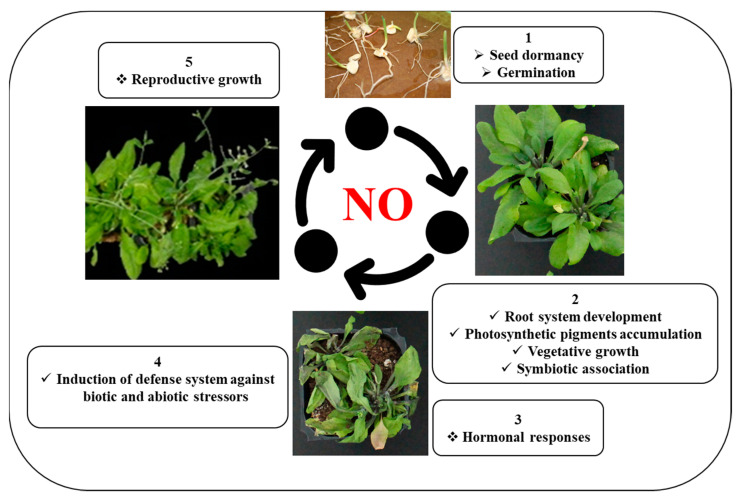
Role of NO in plant growth and development and defense responses against biotic and abiotic stress conditions.

**Table 1 ijms-24-04782-t001:** PTMs induced by NO through S-nitrosation or tyrosine nitration.

Plants	Enzymes	NO-Induced PTMs	References
*Arabidopsis*	3-ketoacyl-Co thiolase 1	S-nitrosation	[97]
Pea and *Arabidopsis*	Hydroxy pyruvate reductase	S-nitrosation/nitration	[76,98]
Pea, *Arabidopsis*, and citrus	Glycolate oxidase	S-nitrosation/nitration	[76,98,99]
*Arabidopsis* and pepper	CAT	S-nitrosation/nitration	[100,101]
Pea	MDAR	S-nitrosation/nitration	[102]
*Arabidopsis*	CuZnSOD	Nitration	[103]
*Arabidopsis*	A total of 53 proteins	S-nitrosation	[97]
*Arabidopsis*	A total of 53 proteins	S-nitrosation	[98]
Different plants	CAT, SOD etc.	S-nitrosation	[31]

## Data Availability

Not Applicable.

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
