# Peer review of "Nitric Oxide Acts as a Key Signaling Molecule in Plant Development under Stressful Conditions"

_ijms, 2023, doi:10.3390/ijms24054782_

Round 1

Reviewer 1 Report

I found this manuscript interesting to read.

But I have some minor revisions as below:

1.       Revise the introduction section. The introduction should be more concise and informative based on nitric oxide related to plant development under stress conditions.

2.       Avoid repeated information in the different sections of the manuscript.

3.       The gene's name (like the NiR gene) should be in italics. Check the other information carefully.  

Author Response

Responses to reviewer 1 comments

We are thankful for the reviewer’s suggestions. We have revised and incorporated all suggested changes accordingly. All changes and improvements in the revised version are done with track changes in the manuscript. 

Reviewer Comment:

Revise the introduction section. The introduction should be more concise and informative based on nitric oxide related to plant development under stress conditions.

Author’s Response:

We are thankful for your kind suggestions. We have thoroughly revised the introduction section of the manuscript by adding information related to NO and plant development under stress conditions.

Reviewer Comment:

Avoid repeated information in the different sections of the manuscript.

Author’s Response:

We have removed all repetitive information in the revised version of our manuscript. Thank you.    

Reviewer Comment:  

The gene's name (like the NiR gene) should be in italics. Check the other information carefully.

Author’s Response:

Thank you so much for your kind suggestions. We have italicized the all genes’ names and revised our manuscript carefully.   

Reviewer 2 Report

Dear authors,

I have received the paper without numbering the lines, that makes the review process much more complicated

*Abstract: “To the best of our knowledge, this is the first review to cover the biosynthesis and signalling of NO as well as its role in plant development and stress mitigation” I'm sorry, but I'm afraid this is not correct, a quick search of the existing bibliography has given me these results and there are many more. This topic is a hot topic, and has already been covered in depth before.

doi: 10.1016/j.niox.2017.12.005

doi: 10.1016/j.niox.2022.11.004

doi: 10.1093/jxb/erz339

doi: 10.1111/ppl.13143

* This case that I explain occurs with several reviews citations. I expose this only as an example. Reference 14 is cited almost 20 times. Which is a review of NO and phytohormones from 2013. Please, correct this and cite the specific works and not one of the reviews that cite them. Do this with all other revisions that are over-cited. Otherwise, what it seems is that they paraphrases those previous reviews.

-Introduction:

*“Arabidopsis plants treated with sodium nitroprusside (SNP)”. Please define the role of the SNP in NO synthesis, the first time it appears

*Page 3, first paragraph: molecule of the year” [21].” But on Page 1, second paragraph it says this: molecule of the year [5]. Which of the two references is correct?

*Page 3, second paragraph: “ antibodies reduced the production of NO [23]”. please indicate which types of antibodies

*Page 3, second paragraph: “Moreover, it also lacks NOS activity can does not help realize oxidation of arginine to form NO in vitro [26, 27]”. Please rewrite this sentence, I don't understand the meaning

* Page 3, third paragraph: “an enzyme called NR, which accelerates…..” accelerate, is not the best verb to explain NR activity. Please rewrite the sentence

* Page 3, third paragraph: “Later, NR activity was also found in other plants, including maize, cucumber, sunflower, and spinach [30, 32]”. Please, rewrite the sentence, NR had been discovered long before, did you mean its role in the synthesis of NO in those organisms, right?

* Page 3, fourth paragraph: ”Since NiR derived from plastids reduces nitrite to ammonia under normal circumstances, the nitrite content of plant cells is significantly lower than their nitrate content”. It needs the correct reference.

* Page 3” Chlamydomonas reinhardtii” Please indicate that it is a microalgae

* End Page 3. “NR can function as a cofactor” Indicate that in Chlamydomonas the NR has also been involved in the detoxification of NO through THB1

* Page 4. “including xanthine oxidase (ZO)” The correct abbreviation is XO. In the figure it is also like ZO, please, change

*Page 4” Numerous enzymes, such as horseradish peroxidase (HPOX), xanthine oxidoreductase (XOR), xanthine dehydrogenase (XDG), cyt P450, polyamine oxidase (PAO), polyam-ine oxidoreductase (POR), and copperamine oxidase (CuAO), also produce NO in plant cells.” the correct references need to be indicated

* Page 4“xanthine dehydrogenase (XDG)” The correct abbreviation is XDH

*Page 4” For example, Bethke et al. discovered that the ectoderms of pasteurized barley cells used a nonenzymatic mechanism to decrease nitrite to NO” what do you mean by decrease? please rewrite

*Legend Figure 1:” nitrite ion (NO3-), nitrate ion (NO2-)” It's the other way around

*Section: “4. Interactions between NO and protein” The title has nothing to do with what is discussed in the entire first paragraph. Why do they put proteins in the title? Change the title to one appropriate to what is being described. In the second paragraph of this section. The first part of the second paragraph has nothing to do with proteins either, you are talking about mRNA

* In table 1. Reference 74 are 62 proteins not 53

*Page 6-end” Some of the important enzymes whose activity can be regulated by NO via PTMs are shown in Table 1.” rewrite is not understandable

* Page 8-second paragraph ” the regulation of this mechanism may be influenced by their likelihood of coming into contact [16]”. Please rewrite is not understandable

*Page 9: “According to Bethke, Libourel, Aoyama, Chung, Still and Jones [42],” please, correct

* Throughout the manuscript the length of the paragraphs is too long, dividing the paragraphs into several will make the reading easier

*Page 9” For example, during GA signaling transduction, hormone molecules connect to GA INSENSITIVE DWARF1 (GID1) receptors, which then bind a DELLA protein and drive the GA-GID1-DELLA complex to the E3 ubiquitin ligase SLEEPY1 (SLY1). This promotes DELLA destruction in the proteasome” The correct reference is needed

*Page 10: “GA share a shared signaling pathway” reword, please

* The second paragraph on page 11 should be completely rewritten. Why is it put there? It has nothing to do with SA. Talk about all the other hormones previously discussed. Place each sentence of that paragraph in the section that corresponds to it.

* Table 2 does not provide more information than what has been previously stated in the text. It's totally expendable, please remove.

*Page 12: “Plants also need specific transporters to absorb minerals; these trans-porters may also be used by other metals without identifiable biological functions” I do not understand the reason for this phrase here? what justification do you have? It has nothing to do with anything in this paragraph.

*Page 13:” In addition, it has been demonstrated that NO plays a role in the regulation of plant defenses against pathogenic fungi” The correct reference is needed.

*Section 6: Role of NO in plant growth and development under normal and stressful conditions. Many of the citations listed in this section have already been covered previously in other sections, please identify, and delete them. It does not contribute anything to repeat them in a different section, leave only the new ones. I will remove this section, it seems that it is only there to justify the title of the paper. The title of the paper should be changed, it does not reflect what the paper is about for the most part.

*Page 13: “When plants are exposed to biotic and abiotic stress conditions, NO induces various plant defense mechanisms [5, 17]” . The same has been said previously.

*Page 13: “NO also significantly contributes to plant reproduction” and where is the reference?

* Figure 2 is incomprehensible, and I do not see what it contributes

Author Response

Responses to reviewer 2 comments

We are thankful for the reviewer’s suggestions. We have revised and incorporated all suggested changes accordingly. All changes and improvements in the revised version are done with track changes in the manuscript. 

Reviewer Comment:

*Abstract: “To the best of our knowledge, this is the first review to cover the biosynthesis and signalling of NO as well as its role in plant development and stress mitigation” I'm sorry, but I'm afraid this is not correct, a quick search of the existing bibliography has given me these results and there are many more. This topic is a hot topic, and has already been covered in depth before.

Author’s Response:

We are thankful for your positive response and kind suggestions. We have revised the statement in the abstract section of the manuscript.

Reviewer Comment:

Reference 14 is cited almost 20 times. Which is a review of NO and phytohormones from 2013. Please, correct this and cite the specific works and not one of the reviews that cite them. Do this with all other revisions that are over-cited. Otherwise, what it seems is that they paraphrases those previous reviews.

Author’s Response:

We are thankful for your kind suggestions all repeated citation has been deleted in the revised manuscript.

Reviewer Comment:  

*“Arabidopsis plants treated with sodium nitroprusside (SNP)”. Please define the role of the SNP in NO synthesis, the first time it appears.

Author’s Response:

We are thankful to you for raising such a meticulous point. We added SNP’s role in NO synthesis in the introduction section.   

Reviewer Comment:  *Page 3, first paragraph: molecule of the year” [21].” But on Page 1, second paragraph it says this: molecule of the year [5]. Which of the two references is correct?

Author’s Response: We have revised the said reference (Levine et al., 2012), and now consistent in the manuscript. Thank you

Reviewer Comment:  *Page 3, second paragraph: “ antibodies reduced the production of NO [23]”. please indicate which types of antibodies

Author’s Response:

Monoclonal and polyclonal antibodies reduce NO production, and we have added this information to the manuscript. Thank you.

Reviewer Comment:  *Page 3, second paragraph: “Moreover, it also lacks NOS activity can does not help realize oxidation of arginine to form NO in vitro [26, 27]”. Please rewrite this sentence, I don't understand the meaning

Author’s Response:

We have removed this statement from the revised manuscript. Thank you.

Reviewer Comment:  * Page 3, third paragraph: “an enzyme called NR, which accelerates…..” accelerate, is not the best verb to explain NR activity. Please rewrite the sentence

Author’s Response:

We have replaced the word “accelerates” with “catalyze”. Thank you.

Reviewer Comment:  * Page 3, third paragraph: “Later, NR activity was also found in other plants, including maize, cucumber, sunflower, and spinach [30, 32]”. Please, rewrite the sentence, NR had been discovered long before, did you mean its role in the synthesis of NO in those organisms, right?

Author’s Response:

We have rewritten the sentence. Thank you.

Reviewer Comment:  * Page 3, fourth paragraph: ”Since NiR derived from plastids reduces nitrite to ammonia under normal circumstances, the nitrite content of plant cells is significantly lower than their nitrate content”. It needs the correct reference.

Author’s Response:

We have added the reference. Thank you.

Reviewer Comment:  * Page 3” Chlamydomonas reinhardtii” Please indicate that it is a microalgae

Author’s Response:

We have added the word “unicellular algae” in the manuscript. Thank you.

Reviewer Comment:  * Page 4. “including xanthine oxidase (ZO)” The correct abbreviation is XO. In the figure it is also like ZO, please, change

Author’s Response:

We have added the word “XO” for xanthine oxidase instead of “ZO”  in the manuscript. Thank you.

Reviewer Comment:  *Page 4” Numerous enzymes, such as horseradish peroxidase (HPOX), xanthine oxidoreductase (XOR), xanthine dehydrogenase (XDG), cyt P450, polyamine oxidase (PAO), polyam-ine oxidoreductase (POR), and copperamine oxidase (CuAO), also produce NO in plant cells.” the correct references need to be indicated

Author’s Response:

We have added the correct references in the manuscript. Thank you.

Reviewer Comment:  * Page 4“xanthine dehydrogenase (XDG)” The correct abbreviation is XDH

Author’s Response:

We have added the word (XDH) instead of (XDG) in the manuscript. Thank you.

Reviewer Comment:  *Page 4” For example, Bethke et al. discovered that the ectoderms of pasteurized barley cells used a nonenzymatic mechanism to decrease nitrite to NO” what do you mean by decrease? please rewrite

Author’s Response:

We have rewritten the sentence. Thank you.

Reviewer Comment:  *Legend Figure 1:” nitrite ion (NO3-), nitrate ion (NO2-)” It's the other way around

Author’s Response:

We corrected it. Thank you.

Reviewer Comment:  *Section: “4. Interactions between NO and protein” The title has nothing to do with what is discussed in the entire first paragraph. Why do they put proteins in the title? Change the title to one appropriate to what is being described. In the second paragraph of this section. The first part of the second paragraph has nothing to do with proteins either, you are talking about mRNA

Author’s Response:

We have corrected the title. Thank you.

Reviewer Comment:  *Page 6-end” Some of the important enzymes whose activity can be regulated by NO via PTMs are shown in Table 1.” rewrite is not understandable

Author’s Response:

We have rewritten the sentence. Thank you.

Reviewer Comment:  * Page 8-second paragraph ” the regulation of this mechanism may be influenced by their likelihood of coming into contact [16]”. Please rewrite is not understandable

Author’s Response:

We have removed this statement from the revised manuscript. Thank you.

Reviewer Comment:  *Page 9: “According to Bethke, Libourel, Aoyama, Chung, Still and Jones [42],” please, correct

Author’s Response:

We have corrected it. Thank you.

Reviewer Comment:  *Page 9” For example, during GA signaling transduction, hormone molecules connect to GA INSENSITIVE DWARF1 (GID1) receptors, which then bind a DELLA protein and drive the GA-GID1-DELLA complex to the E3 ubiquitin ligase SLEEPY1 (SLY1). This promotes DELLA destruction in the proteasome” The correct reference is needed

Author’s Response:

We have added the correct references in the manuscript. Thank you.

Reviewer Comment:  *Page 10: “GA share a shared signaling pathway” reword, please

Author’s Response:

We have rewritten the sentence. Thank you.

Reviewer Comment:  * The second paragraph on page 11 should be completely rewritten. Why is it put there? It has nothing to do with SA. Talk about all the other hormones previously discussed. Place each sentence of that paragraph in the section that corresponds to it.

Author’s Response:

We have deleted this paragraph from the manuscript. Thank you.

Reviewer Comment:  * Table 2 does not provide more information than what has been previously stated in the text. It's totally expendable, please remove.

Author’s Response:

We have deleted Table 2 from the manuscript. Thank you.

Reviewer Comment:  *Page 12: “Plants also need specific transporters to absorb minerals; these trans-porters may also be used by other metals without identifiable biological functions” I do not understand the reason for this phrase here? what justification do you have? It has nothing to do with anything in this paragraph.

Author’s Response:

We have deleted this sentence from the manuscript. Thank you.

Reviewer Comment:  *Page 13:” In addition, it has been demonstrated that NO plays a role in the regulation of plant defenses against pathogenic fungi” The correct reference is needed.

Author’s Response:

We have added the correct references in the manuscript. Thank you.

Reviewer Comment:  *Section 6: Role of NO in plant growth and development under normal and stressful conditions. Many of the citations listed in this section have already been covered previously in other sections, please identify, and delete them. It does not contribute anything to repeat them in a different section, leave only the new ones. I will remove this section, it seems that it is only there to justify the title of the paper. The title of the paper should be changed, it does not reflect what the paper is about for the most part.

Author’s Response:

We have changed the title of the manuscript. Thank you.

Reviewer Comment:  *Page 13: “When plants are exposed to biotic and abiotic stress conditions, NO induces various plant defense mechanisms [5, 17]” . The same has been said previously.

Author’s Response:

We have deleted this sentence from the manuscript. Thank you.

Reviewer Comment:  *Page 13: “NO also significantly contributes to plant reproduction” and where is the reference?

Author’s Response:

We have added the correct references in the manuscript. Thank you.

Reviewer Comment:  * Figure 2 is incomprehensible, and I do not see what it contributes

Author’s Response:

We have added further information to this section of the manuscript. Thank you.

Reviewer 3 Report

This review shows Nitric oxide (NO) biosynthesis, interactions between NO and phytohormones, and role of NO in plant growth and development under normal and stressful conditions. 

There are some comments:

1.       Fig. 2 is blurred, it is recommended to increase the resolution of the image.

2.       About Fig.1, can you explain the meaning of ‘’ in ‘NOS What’s more, it would be better to beautify the picture.

3.       The second paragraph of Part 5.4 has the gene name without italics.

4.       In the row of Table 2 citing [105], there is a single letter ‘P’ in the 'Interaction of NO with hormones', presumably an incomplete expression.

5.       Part 2.1 could ideally be written underneath Part 2 to introduce NO biosynthesis. Then, the content of Part 2.2 belongs to ‘enzymatic and nonenzymatic pathways’, but there is no mention of this in this part, and ‘other enzymatic and nonenzymatic pathways appear directly in Part 2.3. It is proposed that Part 2.2 and 2.3 can be combined.

Author Response

Responses to reviewer 3 comments

We are thankful to you for your positive response and kind suggestions. We have revised and incorporated all suggested changes accordingly. All changes and improvements in the revised version are done with track changes in the manuscript. 

Reviewer Comment: 1. Fig. 2 is blurred, it is recommended to increase the resolution of the image.

Author’s Response: We have increased the resolution of the image. Thank you.

Reviewer Comment: 2. About Fig.1, can you explain the meaning of ‘?’ in ‘NOS?’? What’s more, it would be better to beautify the picture.

Author’s Response: In Figure 1, the sign of interrogation denotes that the mechanism for the production of NOS is not been fully discovered yet in the higher plants. Thank you.

Reviewer Comment: 3. The second paragraph of Part 5.4 has the gene name without italics.

Author’s Response: We have italicized the gene name. Thank you

Reviewer Comment: 4. in the row of Table 2 citing [105], there is a single letter ‘P’ in the 'Interaction of NO with hormones', presumably an incomplete expression.

Author’s Response: We have rectified the single letter “P” which denotes Phosphorous. Thank you

Reviewer Comment: 5. Part 2.1 could ideally be written underneath Part 2 to introduce NO biosynthesis. Then, the content of Part 2.2 belongs to ‘enzymatic and nonenzymatic pathways’, but there is no mention of this in this part, and ‘other enzymatic and nonenzymatic pathways’ appear directly in Part 2.3. It is proposed that Part 2.2 and 2.3 can be combined.

Author’s Response: We totally agree with your meticulous suggestion. We have combined both sections (2.2 and 2.3). Thank you. 

Round 2

Reviewer 2 Report

Dear Authors,

I believe that the authors have responded favorably to all my recommendations and I accept the paper in its current version.